# Observation of interlayer plasmon polaron in graphene/WS₂ heterostructures

Søren Ulstrup [1] ✉, Yann in 't Veld [2], Jill A. Miwa[1], Alfred J. H. Jones [1], Kathleen M. McCreary [3], Jeremy T. Robinson [3], Berend T. Jonker[3], Simranjeet Singh [4], Roland J. Koch [5], Eli Rotenberg [5], Aaron Bostwick [5], Chris Jozwiak [5], Malte Rösner [2] ✉ & Jyoti Katoch [4] ✉

Harnessing electronic excitations involving coherent coupling to bosonic modes is essential for the design and control of emergent phenomena in quantum materials. In situations where charge carriers induce a lattice distortion due to the electron-phonon interaction, the conducting states get "dressed", which leads to the formation of polaronic quasiparticles. The exploration of polaronic effects on low-energy excitations is in its infancy in two-dimensional materials. Here, we present the discovery of an interlayer plasmon polaron in heterostructures composed of graphene on top of single-layer WS₂. By using micro-focused angle-resolved photoemission spectroscopy during in situ doping of the top graphene layer, we observe a strong quasiparticle peak accompanied by several carrier density-dependent shake-off replicas around the single-layer WS₂ conduction band minimum. Our results are explained by an effective many-body model in terms of a coupling between single-layer WS₂ conduction electrons and an interlayer plasmon mode. It is important to take into account the presence of such interlayer collective modes, as they have profound consequences for the electronic and optical properties of heterostructures that are routinely explored in many device architectures involving 2D transition metal dichalcogenides.

Sophisticated heterostructure designs involving two-dimensional (2D) crystals with pre-defined lattice mismatch and interlayer twist angle have emerged as promising platforms for tailoring potential energy surfaces and excitations in solid-state quantum simulators[1,2]. While these systems leverage fine-control of complex lattice structures and quantum states, the close proximity of materials may further induce additional interlayer correlation effects[3]. For example, in heterostructures composed of graphene and semiconducting transition metal dichalcogenides (TMDs), superlattice bands are generated concomitant with screening-induced band shifts that dictate quasiparticle band alignments and gaps[4–7]. Intriguingly, recent experiments on twisted bilayer graphene interfaced with single-layer (SL) WSe₂ point towards even richer interactions, as the presence of SL WSe₂ stabilises superconductivity below the magic twist angle of bilayer graphene[8]. In SL WS₂ contacted with the topological insulator Bi₂Se₃, interlayer exciton-phonon bound states have been detected[9]. Such observations point to the importance of interlayer collective excitations involving bosonic modes. These may lead to the formation of polaronic quasiparticles that dramatically impact charge transport, surface reactivity, thermoelectric,

[1]Department of Physics and Astronomy, Interdisciplinary Nanoscience Center, Aarhus University, 8000 Aarhus C, Denmark. [2]Institute for Molecules and Materials, Radboud University, 6525 AJ Nijmegen, the Netherlands. [3]Naval Research laboratory, Washington, DC 20375, USA. [4]Department of Physics, Carnegie Mellon University, Pittsburgh, PA 15213, USA. [5]Advanced Light Source, E. O. Lawrence Berkeley National Laboratory, Berkeley, CA 94720, USA. ✉e-mail: ulstrup@phys.au.dk; m.roesner@science.ru.nl; jkatoch@andrew.cmu.edu

and optical properties, as observed in a variety of crystals and interfaces composed of polar materials[10–14]. Similarly, when oscillations of the charge density couple to conduction electrons the more elusive plasmon polaron emerges[15], which has been detected in electron-doped semiconductors[16–18] and graphene[19].

We endeavour to determine how the electronic excitation spectrum of a representative semiconducting SL TMD is affected by a doped graphene overlayer, as is present in a variety of device architectures[20–25]. To this end, we focus on SL WS$_2$ as this material exhibits a direct band gap at the $\bar{K}$-point of the Brillouin zone (BZ) and a large spin-orbit coupling (SOC) induced splitting of the valence bands, allowing to simultaneously resolve energy- and momentum-dependent electronic excitations around the valence and conduction band extrema using high-resolution angle-resolved photoemission spectroscopy (ARPES)[26,27]. The heterostructures are supported on 10−30 nm thick hBN, which serves two purposes: (i) it replicates the heterostructures that are typically used in transport and optical measurements, and (ii) provides an atomically flat and inert interface that preserves the salient dispersion of SL WS$_2$, since hybridization is strongly suppressed due to the large band gap of hBN[26]. The entire stack is placed on degenerately-doped TiO$_2$ in order to prevent charging during photoemission. The quasiparticle band structure from the heterostructure is spatially-resolved using micro-focused angle-resolved photoemission spectroscopy (microARPES) during in situ electron doping by depositing potassium atoms on the surface. In order to determine the effect of the graphene overlayer, we measure two types of heterostructures - one with graphene and one without. A schematic of our doped heterostructures is presented in Fig. 1a. Spectra are collected along the $\bar{\Gamma}$-$\bar{Q}$-$\bar{K}$ direction of the SL WS$_2$ BZ, as sketched in Fig. 1b.

## Results

### Electronic structure of doped WS$_2$ and graphene/WS$_2$

Figure 1c presents ARPES spectra of the effect of strong electron-doping on bare WS$_2$ with potassium atoms deposited directly on the surface. Before doping, the expected band structure of SL WS$_2$ is observed with a local valence band maximum (VBM) at $\bar{\Gamma}$ and a global VBM at $\bar{K}$, a total gap larger than 2 eV, and a SOC splitting of 430 meV in the VBM[28] (see left panel of Fig. 1c). At an estimated highest electron density of $(3.0 \pm 0.2) \cdot 10^{13}$ cm$^{-2}$, induced by the adsorbed potassium atoms, the conduction band minimum (CBM) is populated and the shape of the VBM is strongly renormalized, as observed in the right panel of Fig. 1c and previously reported[26]. The direct band gap at $\bar{K}$ is furthermore reduced to $(1.64 \pm 0.02)$ eV (Supplementary Fig. 1), indicating enhanced internal screening. A detailed view of the CBM region in Fig. 1d, reveals the CBM to be relatively broad with an energy distribution curve (EDC) linewidth of $(0.17 \pm 0.02)$ eV and a momentum distribution curve (MDC) width of $(0.29 \pm 0.02)$ Å$^{-1}$ (Supplementary Fig. 2).

These spectra are contrasted with the situation where a graphene layer is placed on top of WS$_2$ in Fig. 1e. In the undoped case shown in the left panel of Fig. 1e, the bands exhibit the same general features as seen in the left panel of Fig. 1c, although they are noticeably sharper and shifted towards the Fermi energy due to the additional screening of the Coulomb interaction by the graphene[5]. Furthermore, a replica of the WS$_2$ local VBM around $\bar{\Gamma}$ is noticeable close to $\bar{Q}$ due to the superlattice formed between graphene and WS$_2$[6]. Upon doping graphene to an electron density of about $(4.8 \pm 0.1) \cdot 10^{13}$ cm$^{-2}$, the SL WS$_2$ valence band shifts down in energy and the shape of the VBM does not renormalize as in the case of bare WS$_2$ (see right panel of Fig. 1e and Supplementary Fig. 1) The strongly doped graphene is accompanied by

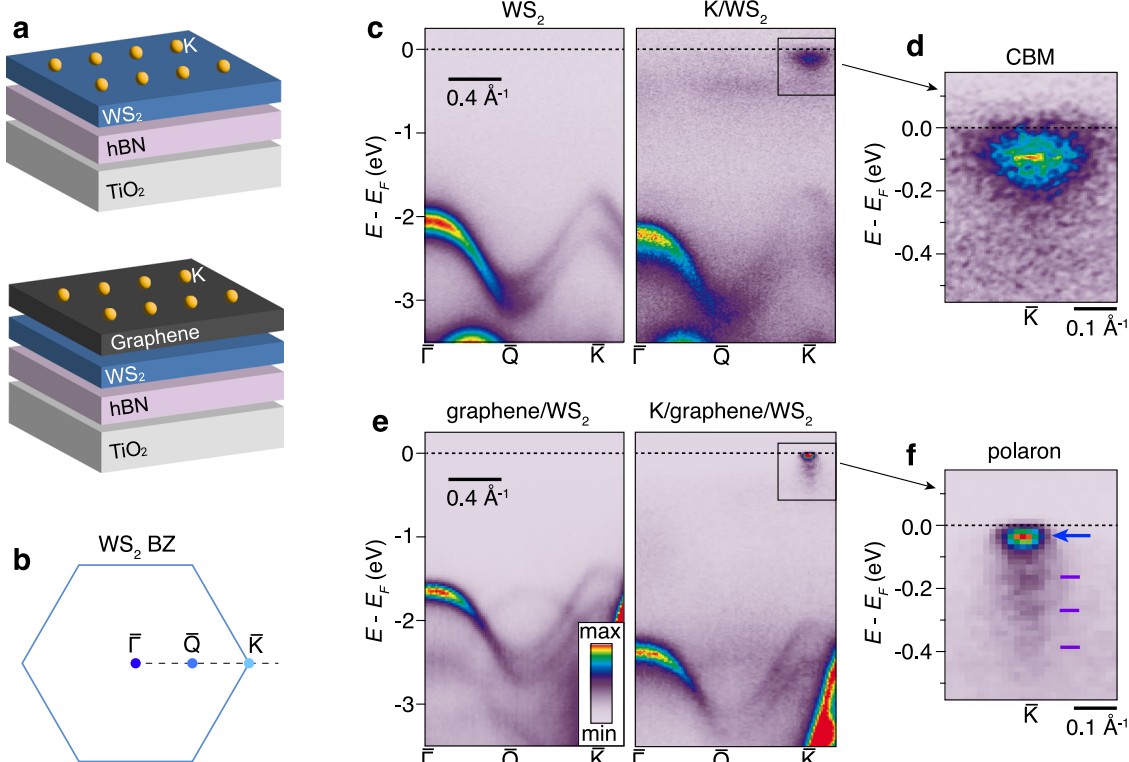

**Fig. 1 | Quasiparticle bands of electron-doped WS$_2$ heterostructures. a** Layout of systems with doping achieved by deposition of potassium atoms. **b** Brillouin zone (BZ) of SL WS$_2$ with ARPES measurement direction marked by a dashed line. **c**, ARPES spectra of bare (left panel) and potassium-dosed WS$_2$ (right panel) supported on hBN. The achieved electron density in the strongly doped case is estimated to be $(3.0 \pm 0.2) \cdot 10^{13}$ cm$^{-2}$. **d** Close-up of the CBM region marked in **c**.

**e**, **f** Corresponding ARPES spectra for WS$_2$ with graphene on top. The achieved electron density in the potassium-dosed graphene layer is $(4.8 \pm 0.1) \cdot 10^{13}$ cm$^{-2}$. The close-up of the CBM region of WS$_2$ in **f** reveals the formation of a polaron via a sharp quasiparticle peak, which is demarcated by a blue arrow, and several shake-off replicas marked by purple ticks.

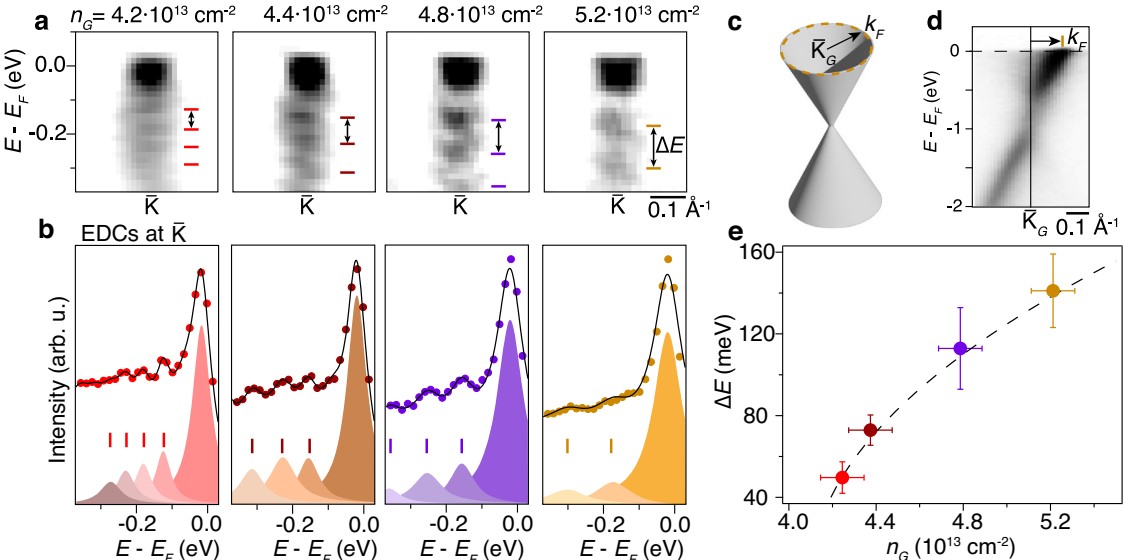

**Fig. 2 | Doping-dependence of shake-off bands. a** Second-derivative ARPES intensity in the CBM region of potassium-dosed graphene/WS$_2$ at the given electron density in graphene ($n_G$). The error bars on the $n_G$ values are $\pm 0.1 \cdot 10^{13}$ cm$^{-2}$. Ticks demarcate shake-off bands and the double-headed arrows indicate their energy separation ($\Delta E$). **b** Energy distribution curves (EDCs) with fits (black curves) to Lorentzian components on a linear background. Peak components are shown with fitted positions marked by colored ticks. **c** Sketch of graphene Dirac cone and Fermi surface (dashed circle) with radius $k_F$ measured simultaneously by ARPES at each doping step. **d** ARPES spectrum of potassium dosed graphene on WS$_2$ with $k_F$ indicated by an arrow. The spectrum is for the maximum achieved doping of graphene of $(5.2 \pm 0.1) \cdot 10^{13}$ cm$^{-2}$. **e** Increase of shake-off energy separation with graphene doping extracted from the analysis. The dashed line represents a fit to a function proportional to $\sqrt{n_G - n_0}$, where $n_0$ is the electron density in graphene that is required to populate the WS$_2$ CBM.

a relatively small occupation in the WS$_2$ CBM (see ARPES spectra of doped WS$_2$ and graphene in Supplementary Fig. 3) The total gap is now $(2.04 \pm 0.02)$ eV (Supplementary Fig. 1) indicating that the non-local Coulomb interaction in WS$_2$ is not fully suppressed. However, the CBM region looks dramatically different, as seen by comparing Fig. 1f and d. In the situation with a doped graphene overlayer, a sharp quasiparticle peak occurs. The peak is accompanied by a series of replica bands towards lower kinetic energy, that are conventionally called shake-off bands. The EDC and MDC linewidths of the main quasiparticle peak are reduced by a factor of 3-4, compared to bare K/WS$_2$ (Supplementary Fig. 2) The feature bears resemblance to a Fröhlich polaron that is observable in ARPES when the conducting electrons couple strongly to phonons[10,11,14,29].

Density functional theory (DFT) calculations for the K/graphene/WS$_2$ heterostructure (see Methods and Supplementary Figs. 4, 5) confirm the experimental results which show that the graphene Dirac bands do not strongly hybridize with the WS$_2$ CBM at $\bar{\text{K}}$, in line with previous reports[30,31]. As a result, there is only a vanishingly small charge transfer from the strongly K-doped graphene layer to the WS$_2$ layer. This explains the experimental observation of strongly doped graphene, accompanied by the small $\bar{\text{K}}$ valley occupation in WS$_2$. This also explains the absence of VBM renormalization in WS$_2$ covered by graphene, as this only occurs at carrier concentrations larger than $(2.0 \pm 0.2) \cdot 10^{13}$ cm$^{-2}$ in WS$_2$[26]. These DFT calculations, however, do not reproduce the still significant band gap or the shake-off bands, pointing towards the important role played here by many-body interactions, that are beyond the scope of DFT calculations.

## Doping-dependence of shake-off bands

In order to understand the origin of the shake-off bands in the dispersion at $\bar{\text{K}}$ in the graphene/WS$_2$ heterostructure, we tune the charge carrier density by sequentially increasing the amount of adsorbed potassium on graphene. After each dosing step, we measure both the WS$_2$ conduction band region and the graphene Dirac cone to correlate the evolution of the shake-off bands spectral line

shapes with the filling of the Dirac cone. Second derivative plots of the ARPES intensity are shown in Fig. 2a to highlight the relatively faint shake-off bands compared to the intense quasiparticle peak for a range of doping where the graphene carrier concentration is varied over a range of $(1.0 \pm 0.1) \cdot 10^{13}$ cm$^{-2}$. Corresponding EDCs with fits to Lorentzian components are shown in Fig. 2b. The graphene wave vector $k_F$, illustrated with the Dirac cone in Fig. 2c, is extracted from ARPES cuts through the center of the graphene Dirac cone at $\bar{\text{K}}_G$, as shown for doped graphene on WS$_2$ in Fig. 2d. The Fermi momentum is then obtained from an MDC fit at $E_F$ and given as the difference in $k$ between the MDC peak position and $\bar{\text{K}}_G$. Note that $\bar{\text{K}}_G$ is determined by mapping the $(E, k_x, k_y)$-dependent ARPES intensity around the Dirac cone. One of the Dirac cone branches is suppressed in Fig. 2d because of strong photoemission matrix element effects along this particular cut, which is taken along the so-called dark corridor[32]. The EDC analysis of the shake-off bands as a function of graphene doping reveals the energy separation between shake-off bands increases from $(50 \pm 8)$ meV to $(141 \pm 18)$ meV and that the increase is proportional to $\sqrt{n_G - n_0}$, as shown in Fig. 2e, while the WS$_2$ CBM binding energy, and thus doping level, approximately stays constant. Note that a minimum carrier density in graphene of $n_0 = (4.1 \pm 0.1) \cdot 10^{13}$ cm$^{-2}$ is required for the WS$_2$ CBM to become occupied and thereby make the shake-off bands observable. The EDC fits in Fig. 2b demonstrate that the shake-off band intensity relative to the main quasiparticle peak diminishes with doping in line with our theoretical analysis below. Combined with the diminishing intensity of shake-offs towards higher binding energies, this reduces the number of shake-off bands we can observe with increasing doping.

These observations provide further clues on the origin of the shake-off bands. An internal coupling between WS$_2$ conducting electrons and phonons can be ruled out because the energy separation of the shake-off bands at high doping exceeds the WS$_2$ phonon bandwidth of 55 meV[33]. Given the significant doping of graphene, there are, however, two other bosonic excitations that could be responsible for the shake-off bands in WS$_2$: phonons and plasmons in graphene. In

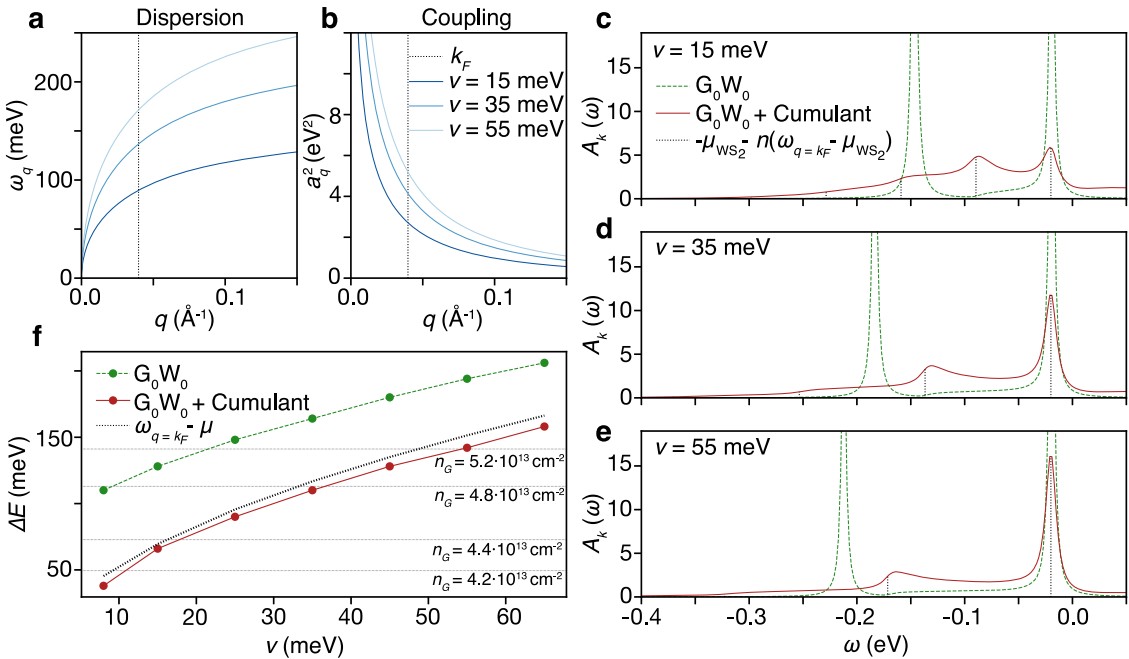

**Fig. 3 | Theoretical results. a, b** The plasmon dispersion $\omega_q$ and electron-plasmon coupling $a_q^2$, respectively, for various $v$. The vertical dotted line denotes $q = k_F$. **c–e** EDCs of the $WS_2$ normal state spectral function in $G_0W_0$ theory (green dashed) and $G_0W_0$+C theory (red solid) at K̄ for a variety of $v$. The vertical dotted black lines denote $\omega = -\mu_{WS_2} - n(\omega_{q=k_F} - \mu_{WS_2})$, for $n = 0$ to 4. **f** Energy splitting $\Delta E$ between the $WS_2$ CBM and the first shake-off band as a function of $v$, in $G_0W_0$ theory (green dashed) and $G_0W_0$+C theory (red solid). The black dotted line denotes $\omega_{q=k_F} - \mu_{WS_2}$, and the gray horizontal lines denote the experimentally measured $\Delta E$.

doped graphene, there are indeed phonons with energies between 150 and 200 meV with significant electron-phonon coupling. These phonon energies change, however, only by up to 20 meV upon tuning the electron doping[34,35] and can thus be ruled out as the origin for the observed shake-offs. In stark contrast, plasmons in 2D materials are known to be significantly affected by the doping level of the system. Indeed, significant plasmon excitations have been observed in graphene in the regime of doping we are considering[19]. Taken together with the significant doping dependence of the energy separation between shake-off bands, this suggests that the observed feature is an interlayer plasmon polaron with unusually sharp line shapes and well-defined shake-offs occurring at moderate $WS_2$ doping levels, unlike the previously observed plasmonic polarons in electron-doped bulk materials[15–17] and in internally doped SL $MoS_2$[18].

**Many-body analysis of electron-plasmon interactions**

To theoretically substantiate this interpretation, we use a generic model consisting of a single layer with a parabolic electronic spectrum, mimicking the occupied $WS_2$ K̄-valley by setting the effective mass to $m^* = 0.3 m_e$ and the chemical potential to $\mu_{WS_2} = 0.02$ eV ($n_{WS_2} \approx 0.5 \cdot 10^{13}$ cm$^{-2}$). As justified by our DFT calculations, we assume that the $WS_2$ and graphene layers are electronically decoupled, such that the only coupling between them is the long-range Coulomb interaction. Based on this, we apply the plasmon pole approximation (PPA) for the screened Coulomb interaction $W_q(\omega)$, where $\omega$ and $q$ are frequency and wavevector, respectively. We subsequently use this formalism within the $G_0W_0$ and retarded $G_0W_0$ + cumulant ($G_0W_0$+C)[36] frameworks to calculate the interacting spectral function within the effective $WS_2$ K̄-valley.

For the plasmon pole model, we assume a 2D plasmon dispersion of the form $\omega_q = \sqrt{4e^2 v q / \varepsilon_q}$, as depicted in Fig. 3a. Here the environmental screening is taken into account using a non-local background dielectric function $\varepsilon_q$, which in the long wavelength limit is given by $\varepsilon_q = \varepsilon_{ext} + qh(\varepsilon_{int}^2 - \varepsilon_{ext}^2)/(2\varepsilon_{ext})$[37], where $\varepsilon_{ext} = 3.0$ and $\varepsilon_{int} = 8.57$ are the

dielectric constants of the substrate and the $WS_2$ layer, respectively, and $h \approx 3.0$ Å an effective dielectric thickness of the $WS_2$ layer. In the plasmon dispersion, $v$ is a tunable parameter which would correspond to a chemical potential in an isolated two-dimensional free electron gas, that here controls the energy scale of the plasmon. The electron-plasmon coupling $a_q^2$ is given by the usual long-wavelength PPA expression $a_q^2 = \omega_q U_q / 2$, with $U_q = 2\pi e^2/(A\varepsilon_q q)$ the background screened Coulomb interaction in the $WS_2$ layer and $A = 8.79$ Å$^2$ the $WS_2$ unit-cell area. In Fig. 3b we show $a_q^2$ for a variety of plasmon energy scales $v$. Note that the electron-plasmon coupling and the plasmon dispersion are related, such that $a_q^2$ increases as $v$ increases.

In Fig. 3c–e we show EDCs of the dressed spectral function $A_k(\omega)$ within the effective $WS_2$ K̄-valley, for various plasmon energy scales $v$. Within both $G_0W_0$ and $G_0W_0$+C theories, we identify the expected CBM quasiparticle peak at $\omega = -0.02$ eV and a plasmon polaron shake-off peak with reduced intensity at lower energies. Within $G_0W_0$+C this is extended to a whole series of partially pronounced plasmon polaron shake-off peaks, which reduce in intensity for peaks further from the CBM. As $v$ is enhanced, the separation between shake-off bands $\Delta E$ increases and the shake-off peak intensity decreases. These results are reminiscent of polarons formed by dispersionless bosons, where the energy separation between shake-off bands is given by the boson frequency $\omega_b$[38]. This suggests that, even though the 2D plasmon is a highly dispersive mode, there exists an effective plasmon frequency which dictates the energy separation $\Delta E$. Since $WS_2$ is only weakly doped we can evaluate the spectral function of the first shake-off band in $G_0W_0$+C theory analytically (see Methods) and understand that the shake-off bands appear in multiples of $\omega_{q=k_F} - \mu_{WS_2}$ below the CBM (indicated by vertical black lines in Fig. 3c–e), with $k_F \approx 0.04$ Å$^{-1}$ the $WS_2$ Fermi wavevector. To confirm this prediction, we plot in Fig. 3f the energy splitting $\Delta E$ in $G_0W_0$+C theory (red line) as a function of the plasmon energy scale $v$, which follows the analytically predicted $\Delta E = \omega_{q=k_F} - \mu_{WS_2}$ (dotted line). From the analytical derivations we also

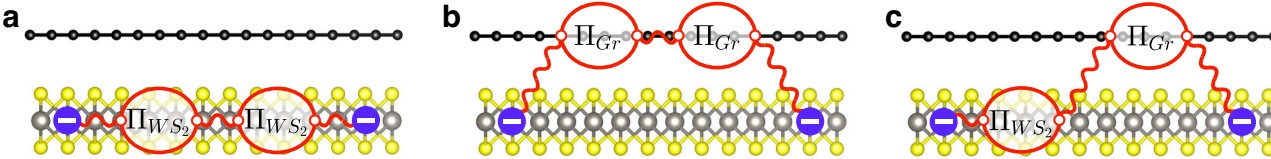

**Fig. 4 | Illustrations of the Coulomb interaction in WS₂ and its screening channels in graphene/WS₂ heterostructures.** Wavy lines and "bubbles" represent bare Coulomb interactions and polarization processes, respectively. **a** Coulomb interaction and screening from WS₂ only. **b** Coulomb interaction between electrons in WS₂ screened by graphene polarization processes only, which couple graphene plasmons to the WS₂ Coulomb interaction. **c**, Illustration of mixed screening channels from WS₂ and graphene. Interlayer polarization effects are suppressed due to the vanishingly small hybridization between the WS₂ $\bar{K}$ valley and graphene's Dirac cone.

understand that the intensity of the first shake-off peak is proportional to

$$A_{k=\bar{K}}(\omega = -\omega_{q=k_F}) \propto \frac{a_{q=k_F}^2}{(\omega_{q=k_F} - \mu_{WS_2})^2} \frac{v_F}{|v_{pl} - v_F|}, \tag{1}$$

with $v_{pl} = \partial\omega_q/\partial q|_{q=k_F}$ the plasmon group velocity at $q = k_F$ and $v_F$ the WS₂ Fermi velocity. Due to the low WS₂ occupation, both $\mu_{WS_2} < \omega_{q=k_F}$ and $v_F < v_{pl}$, which explains the reduced intensity of the shake-off peaks upon enhancing the plasmon energy scale $v$. Finally, the analytic $G_0W_0+C$ expressions explain that the non-zero intensity between the shake-off peaks and the CBM is a consequence of the gapless dispersion of the 2D plasmon mode.

Comparing $G_0W_0$ and $G_0W_0+C$ theory, we show in Fig. 3c–e that the EDCs predicted by $G_0W_0$ theory (green lines) capture only a single shake-off band, whereas $G_0W_0+C$ theory (red lines) predicts an infinite series of shake-off bands. Furthermore, Fig. 3f shows that $\Delta E$ predicted by $G_0W_0$ theory overestimates $\Delta E$ from $G_0W_0+C$ theory by more than 50 meV for all plasmon energy scales $v$ considered. These discrepancies are consistent with earlier works[15,18,36,38,39] and are a clear sign that correlations beyond $G_0W_0$ theory (i.e., vertex corrections) are playing a significant role here.

From the analysis above, we understand that in order to observe an enhancement of $\Delta E$ on the order of 100 meV upon K-adsorption, the plasmon energy at $q = k_F$ should increase by the same amount. Additionally, the group velocity of the plasmon should be of similar magnitude to the WS₂ Fermi velocity to increase the shake-off intensity. These restrictions allow us to investigate the origin of the relevant plasmon mode. To this end, we depict in Fig. 4 the three possible screening channels to the Coulomb interaction within the WS₂ layer, which could be responsible for the relevant plasmonic mode. Figure 4a describes screening processes from within the WS₂ layer, which induces a plasmon mode that is spatially restricted to the WS₂ layer. Due to the quadratic dispersion of the WS₂ CBM, this plasmon mode behaves as $\omega_q^{WS_2} = \sqrt{4e^2\mu_{WS_2}q/\varepsilon_q}$ in the long wavelength limit[40]. There are therefore two ways in which the energy of this mode can be tuned: doping of the WS₂ layer and external screening to it. As for doping, from the ARPES data we learn that the WS₂ CBM does not exhibit an observable shift over the range of K-doping where the polaron effect emerges. Additionally, no shifts in the valence bands are observed, such that we can conclude that the WS₂ occupation is not significantly altered over this doping range. We can therefore exclude that WS₂ doping significantly changes the WS₂ plasmon energy. As for screening, static screening from the graphene layer can change the energy scale of the WS₂ plasmon and is sensitive to the doping of graphene. However, within a Thomas-Fermi screening model, we understand that as the doping of graphene is increased, the screening increases, such that the WS₂ plasmon energy decreases with enhanced K-doping. This is opposite to the trend that is observed experimentally, thereby excluding this mechanism. We conclude that the WS₂ plasmon energy is not significantly enhanced upon K-doping, which means it cannot

cause changes in the shake-off energy splitting on the order of 100 meV.

Figure 4b describes a dynamical screening process from the graphene layer, which induces a graphene-like plasmon mode which is coupled into the WS₂ layer via long-range Coulomb interaction. The experimental data as well as the DFT results show that the graphene layer is readily doped by K-adsorption, such that this plasmon mode, which behaves as $\omega_q^G = \sqrt{2e^2\mu_G q/\varepsilon_q}$ in the long wavelength limit[41], significantly increases in energy. While the trends in this scenario are correct, the graphene plasmon energy scale of $\omega_{q=k_F}^G \approx 460$ meV at the measured graphene occupation of $n_G = 4.8 \cdot 10^{13}$ cm⁻² yields an energy separation $\Delta E$ which is too large compared to the measured value of $(113 \pm 20)$ meV. In addition, the group velocity of the graphene plasmon $v_{q=k_F}^{pl,G}$ is approximately 4 times larger than the WS₂ $v_F$, such that the intensity of the resulting shake-off peak is reduced. However, hybridization with another boson mode, such as a phonon mode in graphene, could flatten the plasmon dispersion and lower its energy at $q = k_F$ to a more suitable regime, such that it could induce the observed plasmon polaron bands in the WS₂ layer.

Finally, Fig. 4c describes interlayer screening processes, which induce interlayer plasmon modes. These can be interpreted as hybridized graphene and WS₂ plasmon modes. Such modes live on energy scales in between those of decoupled intralayer graphene and WS₂ modes, while at the same time being sensitive to the graphene occupation. These hybridized interlayer plasmon modes can explain all relevant experimental observations without the need of taking further bosonic excitations into account.

Based on this, we conclude that the shake-off bands observed in K-doped graphene/WS₂ heterostructures are signatures of interlayer plasmon polarons, which are formed by WS₂ electrons coupling either to renormalized graphene plasmon modes, or to interlayer hybridized plasmon modes as a result of the inter-layer Coulomb interaction in the heterostructure.

## Discussion

Taking only the WS₂ layer in the passive screening and/or doping background of K-doped graphene into account cannot explain the experimentally observed K-tunable formation of a series of shake-off bands within the WS₂ $\bar{K}$-valley. Our results thus clearly underline the relevance of the full heterostructure, and especially the interlayer Coulomb coupling, in facilitating the formation of plasmon polaron bands in the WS₂ layer. The graphene layer acts as a buffer to weaken the doping of the WS₂ layer, as well as providing an interlayer plasmon mode, which couples strongly to the WS₂ electrons and leads to the formation of plasmon polarons. The sensitivity of these interlayer plasmon modes to the graphene occupation leads to a high degree of tunability in the positions of the plasmon polaron shake-off bands. The missing higher-order shake-off bands in the $G_0W_0$ approximation are further evidence of the need for vertex corrections[18,38,42], which we incorporated here within the $G_0W_0+C$ approach.

The impact of these findings could be far-reaching, as interfaces between graphene and TMDs have been exploited in various ways: to induce large spin-orbital proximity effects[43], for the stabilization of superconductivity below magic angle twists in bilayer graphene interfaced with $WSe_2$[8], or for charge carrier control of Wigner crystallization and realizations of discrete Mott states in dual-gated TMD heterobilayers contacted with graphite[1,2]. Our observation of interlayer polaronic quasiparticles induced by interlayer Coulomb coupling and upon adding charge to a contacting graphene layer will thus be important to consider in the interpretation and modelling of device measurements. Further experiments will be required to evaluate their impact on the optoelectronic properties and band engineering of heterostructures as well as their utility for ultrathin photonics and plasmonic devices.

## Methods

### Fabrication of heterostructures

First, bulk hBN crystals (commercial crystal from HQ Graphene) were exfoliated onto 0.5 wt% Nb-doped rutile $TiO_2(100)$ substrate (Shinkosha Co., Ltd) using scotch tape to obtain 10-30 nm thick hBN flakes. Next, we transferred chemical vapor deposition (CVD) grown SL $WS_2$ onto a selected thin hBN flake using a thin polycarbonate film on a polydimethylsiloxane stamp using a custom-built transfer tool. This was followed by the transfer of CVD graphene on top of the $WS_2$/hBN stack[6]. After each transfer process, the sample surface was cleaned by annealing in ultrahigh vacuum (UHV) at 150 °C for 15 min to remove any unwanted residues or adsorbates from the surface.

### Micro-focused angle-resolved photoemission spectroscopy

The photoemission experiments were carried out in the microARPES end-station of the MAESTRO facility at the Advanced Light Source. Samples were transported through air and given a 1 hour anneal at 500 K in the end-station prior to measurements. The base pressure of the system was better than $5 \cdot 10^{-11}$ mbar and the samples were kept at a temperature of 78 K throughout the measurements.

Energy- and momentum-resolved photoemission spectra were measured using a Scienta R4000 hemispherical electron analyser with custom-made deflectors. All samples were aligned with the $\bar{\Gamma} - \bar{K}$ direction of the $WS_2$ Brillouin zone (BZ) aligned along the slit of the analyser. Measurements on $WS_2$ samples without a graphene overlayer were performed with a photon energy of 145 eV, while measurements on samples with a graphene overlayer were done with a photon energy of 80 eV. These energies were chosen on the basis of photon energy scans revealing the optimum matrix elements for clearly resolving the $WS_2$ and graphene band structures. The photon beam was focused to a spot-size with a lateral diameter of approximately 10 $\mu$m using Kirkpatrick-Baez (KB) mirrors.

Electron-doping of samples was achieved by depositing potassium (K) from a SAES getter source in situ. Each dose had a duration of 40 s. After each dose, the $\bar{\Gamma} - \bar{K}$ cut of $WS_2$ was acquired for 5 minutes followed by a measurement around the Dirac point of graphene for 3 minutes. Efficient switching between these two cut directions was achieved using the deflector capability of the analyser, such that all measurements could be done with the sample position held fixed. In $WS_2$ without a graphene overlayer, the carrier concentration in $WS_2$ was estimated using the Luttinger theorem via the Fermi surface area enclosed by the $WS_2$ conduction band. In the samples with a graphene overlayer, we determined the doping of graphene by directly measuring $k_F$, as shown in Fig. 2c, d, and using the relation $n_G = k_F^2/\pi$. It is not possible to determine the doping of $WS_2$ under graphene in a similar way as for bare $WS_2$ because the CBM remains flat and pinned at $E_F$, preventing any meaningful extraction of a Luttinger area. We therefore only report the graphene doping level for graphene/$WS_2$ heterostructures, which can be reliably extracted as described above.

The second derivative plots of the ARPES intensity shown in Fig. 2a of the main paper were obtained using the method described in ref. 44 and merely used as a tool to visualize the data. Analysis of energy and momentum distribution curves was always performed on the raw ARPES intensity.

A total of 3 samples were studied, which were a bare $WS_2$ and two graphene/$WS_2$ heterostructures on separate $TiO_2$ wafers such that fresh doping experiments could be performed on all samples. The two graphene/$WS_2$ heterostructures exhibited twist angles of $(7.5 \pm 0.3)°$ and $(18.1 \pm 0.3)°$ between graphene and $WS_2$, as determined from the BZ orientations in the ARPES measurements. We found identical behaviors with doping and the formation of polarons in the two heterostructures, confirming the reproducibility of our results.

### Density functional theory calculations

To study the hybridization and the possible charge transfer between the graphene and $WS_2$ layers, we performed density functional theory (DFT) calculations using a $4 \times 4$ $WS_2$/$5 \times 5$ graphene supercell with K doping, as indicated in Supplementary Fig. 4. The supercell height has been fixed to about 26 Å to suppress unwanted wavefunction overlap between adjacent supercells. The $WS_2$ lattice constant has been fixed to its experimental value of 3.184 Å while the graphene lattice constant has been strained by about 3% to 2.547 Å to obtain a commensurable heterostructure. The graphene-$WS_2$ interlayer separation has been set to previously reported 3.44 Å[45] and the K-graphene distance has been optimized in DFT yielding 2.642 Å in the out-of-plane direction. All calculations were performed within the Vienna Ab initio Simulation Package (VASP)[46,47] utilizing the projector-augmented wave (PAW)[48,49] formalism within the PBE[50] generalized-gradient approximation (GGA) using $12 \times 12 \times 1k$ point grids and an energy cut-off of 400 eV.

In Supplementary Fig. 5, we show the resulting unfolded band structure (without SOC effects) together with the pristine $WS_2$ band structure (including SOC effects) following the approach from ref. 51 as implemented in ref. 52. From this, we can clearly see that in the heterostructure new states in the gap of $WS_2$ arise, which we identify as graphene bands. Due to unfolding (matrix element) effects, the second linear band forming graphene's Dirac cone is not visible. Upon unfolding to the primitive graphene structure, the Dirac point becomes visible (right panel) showing a graphene Fermi energy of about 0.6 eV in good agreement with the experimentally achieved range. In the upmost valence states around the $\bar{K}$-point, we see that the graphene and $WS_2$ bands hybridize similarly to reported band structures on undoped graphene/$WS_2$[31,45]. In the conduction band region, we however see that graphene states are far from the $\bar{K}$-valley, such that hybridization between graphene $p_z$ and W $d_{z^2}$ orbitals (which are dominating the $\bar{K}$-valley) is almost completely suppressed. As a result, there is negligible charge transfer from graphene to $WS_2$, so that primarily graphene is doped by potassium. This is fully in line with our experimental results.

### Analytical $G_0W_0$+C expressions

For the $G_0W_0$+C calculations, we use the formalism proposed by Kas et al.[36], which is based on the cumulant ansatz for the dressed Green's function $G_\mathbf{k}(t) = G_\mathbf{k}^{(0)}(t)e^{C_\mathbf{k}(t)}$ with the cumulant function given by

$$C_\mathbf{k}(t) = \int d\omega \frac{\beta_\mathbf{k}(\omega)}{\omega^2}\left(e^{-i\omega t} + i\omega t - 1\right), \quad (2)$$

where $\beta_\mathbf{k}(\omega) = |\text{Im}\left(\Sigma_\mathbf{k}^{\text{dyn}}(\omega + \epsilon_\mathbf{k} - \mu)\right)|/\pi$ and $\varepsilon_\mathbf{k} = k^2/(2m^*)$ is the electron dispersion. The self-energy $\Sigma_\mathbf{k}(\omega) = \Sigma_\mathbf{k}^{\text{stat}} + \Sigma_\mathbf{k}^{\text{dyn}}(\omega)$ is the $G_0W_0$ self-energy, which we split into a sum of static and dynamic contributions. The dynamic part can be written in the plasmon pole approximation as

$$\Sigma_\mathbf{k}^{\text{dyn}}(\omega) = \sum_\mathbf{q} a_\mathbf{q}^2 \left(\frac{n_B(\omega_\mathbf{q}) + n_F(\varepsilon_{\mathbf{k}+\mathbf{q}} - \mu)}{\omega - \varepsilon_{\mathbf{k}+\mathbf{q}} + \mu + \omega_\mathbf{q} + i\delta} + \frac{n_B(\omega_\mathbf{q}) + 1 - n_F(\varepsilon_{\mathbf{k}+\mathbf{q}} - \mu)}{\omega - \varepsilon_{\mathbf{k}+\mathbf{q}} + \mu - \omega_\mathbf{q} + i\delta}\right),$$

$$(3)$$

where $n_B$ and $n_F$ are the Bose-Einstein and Fermi-Dirac distributions. Numerical evaluations are done using the expressions above, but for the analytical analysis it is convenient to write the cumulant function as a sum of three terms $C_{\mathbf{k}}(t) = O_{\mathbf{k}}(t) + i\Delta_{\mathbf{k}}t - a_{\mathbf{k}}$, where $O_{\mathbf{k}}(t) = \int d\omega \beta_{\mathbf{k}}(\omega) e^{-i\omega t}/\omega^2$, $\Delta_{\mathbf{k}} = \int d\omega \beta_{\mathbf{k}}(\omega)/\omega$ and $a_{\mathbf{k}} = \int d\omega \beta_{\mathbf{k}}(\omega)/\omega^2$. In this way, the dressed Green's function can be written as

$$G_{\mathbf{k}}(t) = -iZ_{\mathbf{k}}\Theta(t>0)e^{i(-\varepsilon_{\mathbf{k}} + \mu - \Sigma_{\mathbf{k}}^{\text{stat}} + \Delta_{\mathbf{k}} + i\delta)t}e^{O_{\mathbf{k}}(t)}, \qquad (4)$$

with $Z_{\mathbf{k}} = \exp(-a_{\mathbf{k}})$ the renormalization constant. Since we are interested in occupied states, we will neglect the second term of the $G_0W_0$ self-energy in Eq. (3). We will focus on the effective K-valley of the $WS_2$ layer by setting $\mathbf{k} = 0$ and we will assume zero temperature for simplicity. Taking the limit $\delta \to 0$ we find $\beta_{\mathbf{k}=0}(\omega) = \sum_{\mathbf{q}} a_{\mathbf{q}}^2 \Theta(\mu - \varepsilon_{\mathbf{q}})\delta(\omega - \varepsilon_{\mathbf{q}} + \omega_{\mathbf{q}})$, with $\Theta(x)$ the Heaviside step function. Substituting $\beta_{\mathbf{k}=0}(\omega)$ into the three terms of the cumulant function gives

$$O_{\mathbf{k}=0}(t) = \sum_{\mathbf{q}} a_{\mathbf{q}}^2 \frac{e^{-i(\varepsilon_{\mathbf{q}} - \omega_{\mathbf{q}})t}}{(\varepsilon_{\mathbf{q}} - \omega_{\mathbf{q}})^2}\Theta(\mu - \varepsilon_{\mathbf{q}}), \qquad (5)$$

$$\Delta_{\mathbf{k}=0} = \sum_{\mathbf{q}} a_{\mathbf{q}}^2 \frac{1}{\varepsilon_{\mathbf{q}} - \omega_{\mathbf{q}}}\Theta(\mu - \varepsilon_{\mathbf{q}}), \qquad (6)$$

$$a_{\mathbf{k}=0} = \sum_{\mathbf{q}} a_{\mathbf{q}}^2 \frac{1}{(\varepsilon_{\mathbf{q}} - \omega_{\mathbf{q}})^2}\Theta(\mu - \varepsilon_{\mathbf{q}}). \qquad (7)$$

To obtain a Green's function for each shake-off band separately, we expand in Eq. (4) $\exp(O_{\mathbf{k}}(t)) = \sum_n O_{\mathbf{k}}^n(t)/n!$, such that each term in the expansion corresponds to the $n$-th shake-off band. Fourier transforming and subsequently evaluating the spectral function $A_{\mathbf{k}=0}(\omega) = \lim_{\delta \to 0} -\text{Im}(G_{\mathbf{k}=0}(\omega))/\pi$ gives

$$
\begin{aligned}
A_{\mathbf{k}=0}(\omega) = &Z_{\mathbf{k}=0}\delta(\omega + E_{\text{CBM}}) \\
&+ Z_{\mathbf{k}=0}\sum_{\mathbf{q}} a_{\mathbf{q}}^2 \frac{1}{(\varepsilon_{\mathbf{q}} - \omega_{\mathbf{q}})^2}\Theta(\mu - \varepsilon_{\mathbf{q}})\delta\left(\omega + E_{\text{CBM}} - \varepsilon_{\mathbf{q}} + \omega_{\mathbf{q}}\right) \\
&+ \mathcal{O}(O^2),
\end{aligned}
$$
$$(8)$$

where $E_{\text{CBM}}$ is the energy of the CBM. For all parameter regimes considered, $\omega_{\mathbf{q}} - \varepsilon_{\mathbf{q}}$ is a monotonically increasing function of the norm $q$ in the range $0 < q < k_F$. As a consequence, the step-function restricts the shake-off band from a dispersive 2D plasmon mode to the full energy range between $\omega = -E_{\text{CBM}}$ and $\omega = -E_{\text{CBM}} + \mu - \omega_{q=k_F}$, where we used that $\omega_{q=0} = 0$ for 2D plasmons, leading to the maximal energy splitting $\Delta E = \omega_{q=k_F} - \mu$. In contrast, a dispersionless boson mode with energy $\omega_b$ has a smaller allowed energy range $-E_{\text{CBM}} - \omega_b < \omega < -E_{\text{CBM}} + \mu - \omega_b$, which leads to a shake-off feature which is completely detached from the CBM.

At each $\omega$, the spectral intensity of the first occupied shake-off band can be evaluated by approximating $\sum_{\mathbf{q}} f(q) \approx \frac{A}{2\pi}\int qf(q)dq$, with $A$ the unit-cell area, and using the property $\delta(g(x)) = \sum_i \delta(x - x_i)/|g'(x_i)|$ with $x_i$ the solutions of $g(x_i) = 0$. This finally yields

$$A_{\mathbf{k}=0}^{(1)}(\omega) = Z_{\mathbf{k}=0}\frac{A}{2\pi}\frac{a_{q(\omega)}^2}{(\varepsilon_{q(\omega)} - \omega_{q(\omega)})^2}\frac{q(\omega)}{\left|\frac{\partial \omega_q}{\partial q} - \frac{\partial \varepsilon_q}{\partial q}\right|_{q=q(\omega)}}\Theta(0 < q(\omega) < k_F). \qquad (9)$$

with $q(\omega)$ the solution of $\omega + E_{\text{CBM}} = \varepsilon_{q(\omega)} - \omega_{q(\omega)}$. Evaluating this function at the lower edge of the allowed frequency range (i.e., at $q(\omega) = k_F$) yields Eq. (1) of the main text.

## Data availability

The data that support the plots within this paper and other findings of this study are available from the corresponding authors upon request.

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

## Acknowledgements

Y.V. and M.R. thank G. Ganzevoort for useful discussions. J.K. acknowledges funding from the U.S. Department Office of Science, Office of Basic Sciences, of the U.S. Department of Energy under Award No. DE-SC0020323 as-well-as partial support by the Center for Emergent Materials, an NSF MRSEC, under award number DMR-2011876. S.U. acknowledges funding from the Danish Council for Independent Research, Natural Sciences under the Sapere Aude program (Grant No. DFF-9064-00057B) and from the Novo Nordisk Foundation (Grant NNF22OC0079960). J.A.M acknowledges funding from the Danish Council for Independent Research, Natural Sciences under the Sapere Aude program (Grant No. DFF-6108-00409) and the Aarhus University Research Foundation. S.S. acknowledges the support from the National Science Foundation under grant DMR-2210510 and the Center for Emergent Materials, an NSF MRSEC, under award number DMR-2011876. K.M.M., J.T.R., and B.T.J. acknowledge support from core programs at the Naval Research Laboratory. Y.V. and M.R. acknowledge support from the Dutch Research Council (NWO) via the "TOPCORE" consortium. M.R. acknowledges partial support by the European Commission's Horizon 2020 RISE program Hydrotronics (Grant No. 873028). This research used resources of the Advanced Light Source, which is a DOE Office of Science User Facility under Contract No. DE-AC02-05CH11231.

## Author contributions

S.U., M.R. and J.K. designed the research project. K.M.M. and B.T.J. synthesized $WS_2$ and J.T.R. synthesized graphene. S.S. and J.K. assembled the heterostructures. S.U., J.A.M., R.J.K., E.R., A.B., C.J. and J.K. performed the ARPES experiments. S.U., J.A.M., A.J.H.J. and J.K. analyzed the ARPES data. Y.V. and M.R. performed DFT and $G_0W_0$+C calculations. S.U., Y.V., M.R. and J.K. wrote the paper with inputs from all authors.

## Competing interests

The authors declare no competing interests.
