## [Peer Review File · Nature Communications]

Observation of interlayer plasmon polaron in graphene/WS2 heterostructuresREVIEWER COMMENTS

Reviewer #1 (Remarks to the Author):

The authors conducted a combined experimental and theoretical investigation of the electronic and quasiparticle excitations in highly doped graphene/WS₂ heterostructures. Experiments are based on micro- focused angle-resolved photoemission spectroscopy (microARPES), whereas simulations are based on a model for the graphene WS₂ hetero structure.

The main finding of this manuscript consists in the discovery of a polaronic series in the conduction band bottom of monolayer WS₂ due to coupling with plasmonic excitations within graphene. The emergence of plasmonic polarons has been observed in the past in bulk and 2D semiconductors at sufficiently high doping concentration. However, to best of my knowledge this is the first instance for the observation of such phenomena in a hybrid heterostructure. Overall, this is a striking example of hybrid quasiparticle arising from many-body interactions involving electrons and plasmons at different ends of an hetero structure.

These results are based on high quality microARPES measurements. The authors have conducted a scan of different doping concentrations and revealed an increase of satellite energy with doping density. This analysis is the smoking-gun evidence for the plasmonic character of the satellite shake-off peaks. The theoretical analysis is based on the GW approximation for a simple model of the WS₂/graphene interface. This approach is below the state of the art, it describes a single plasmon satellites and it overestimates its energy. Even if the approach suffices to captures the key qualitative features of the experiments and to corroborate the interpretation of plasmon satellites, I believe that it would strengthen the manuscript to resort to a higher level theory, for instance by including vertex correction (as mentioned in the manuscript).

Overall the findings have sufficient novelty and originality to grant

publication in Nature Communications. Before recommending the manuscript for publications, however, there is a final issue that the authors should address, to corroborate their interpretation of experimental data:

The authors attribute plasmons to the graphene layer. However, I would have expected a simpler and more intuitive picture, in which plasmons in the WS₂ layer are screened by graphene. Can the authors exclude this picture and unambiguously attribute plasmonic excitations to the graphene layer? A discussion of this aspect should be included in the manuscript.

Reviewer #2 (Remarks to the Author):

Ulstrup and coauthors investigate the presence of interlayer plasmon polarons in graphene/WS₂ heterostructures. In particular, the authors observe so-called shake-off bands in the ARPES signal of WS₂ conduction band electrons. The qualitative agreement between the measured doping dependence of the shake-off bands and the theoretical model indicates that the observed signatures originate from WS₂ conduction band electrons coupled to graphene plasmons. The findings are clearly presented and the conclusions are well supported. Overall, the work provides an important step forward in the context of heterostructures formed by graphene and transition metal dichalcogenides, which are currently relevant for fundamental studies and potential applications. While I think this work should be published in Nature Communications, I ask the authors to address my comments before I can make a definitive recommendation.

1. The lack of hybridization between the Dirac cone and the WS₂ CBM at K, which is a key aspect for the realization of the interlayer plasmon polaron, was already reported in other works [PRL 127, 276401 (2021); 2D Mater. 10 (2023) 035025]. The authors are aware of at least one of these articles, which they cite in the supplementary information but not in the main manuscript. I ask the authors to refer to these works in the manuscript.

2. The authors consider the structure hBN/WS₂/graphene/dopants. Out of curiosity, how are the results expected to change if the dopants are located on top of WS₂, e.g. in an

hBN/graphene/WS2/dopants structure?

3. It is not clear from Fig.1e that the K valley occupation in WS2 is small, as the authors state. It makes sense that it should be small, but I do not see how the authors reach the conclusion from Fig.1e alone. This should be better explained. It could be helpful, or at least improve transparency, to show in the extended data the conduction band signal of both the WS2 K valley and the graphene Dirac cone together in a single plot.

4. While Fig.2c is very clear, its connection with Fig.2d is not really so, given that only one line and not the cone is resolved in Fig.2d. Why is only one line seen in Fig.2d? And how can the Fermi momentum be extracted from this, i.e. how is the center of the cone extracted from the data? This needs to be more clearly explained.

5. I have a few questions regarding the shake-off bands. Addressing these points in the manuscript or supplementary information could be helpful for future works trying to understand this phenomenon better.

1. Does the number of shake-off bands change with doping?

2. Is the dispersion of the shake-off bands expected to be the same as that of the CBM?

3. Why does the energy and momentum width of the CBM signal decrease with the presence of graphene, and is this reproduced by the theoretical model? Related to this, can an effective mass and temperature be extracted from the energy and momentum width of the CBM signal?

4. Why does the shake-off band intensity relative to the main quasiparticle peak diminish with doping? Perhaps the theoretical model can provide insights into this.

5. I understand that the theory considered can only predict a single shake-off band.

However, I wonder if it is expected that additional sidebands will display the same energy separation as seen in Fig.2b. Related to this, what is the physical meaning of the energetic separation between the shake-off bands? For phonon polarons it is given by the phonon energy, if I am not mistaken. Can it be related to some characteristic energy of the graphene plasmons here?

6. Regarding the sentence “the doping dependence of the line shape of the shake-off bands

is consistent with graphene plasmon excitations coupling to the WS₂ conduction electrons”, I ask the authors to be more precise: how does the line shape depend on doping and how is this consistent with the mentioned effect?

7. Where is the plasmon polaron band in Fig4c? The signatures below the dotted lines are a bit complicated, resembling a sort of triangular shape. Is the polaron band the upper part of this “triangle? Then what are the sides? Please explain more clearly and indicate where the polaron band is in the plot. Indicating the energy separation between the dotted lines and the plasmon polaron band in the plot would help.

8. Can the authors show that the theoretical trend shown in Fig.4e also follows a square root dependence as they state in the text?

9. Can the authors foresee some of the expected consequences of the observed interlayer plasmon polarons? In particular, would they weaken or enhance the electron-electron interactions in WS₂? From this one could qualitatively predict how the stability of correlated many-body phases such as Wigner crystals would be affected.

Reviewer #3 (Remarks to the Author):

The manuscript by Ulstrup et al report the observation of a new many body quasiparticle - an interlayer plasmon polaron. Plasmon polarons have been reported previously, but the observation of the coupling occurring across two different monolayer materials is a significant new discovery. There is much interest in low-energy bosonic modes, such as plasmon polarons, for realising and/or stabilising different strongly correlated electron phases (eg superconductivity) and novel quantum phases. In this particular case, the ability to couple the plasmon in electron doped graphene to the bandstructure of a monolayer semiconductor provides a new avenue for controlling the properties of such materials.

The main experimental evidence is the ARPES data showing shake off bands below the WS₂ conduction band, with energy splitting that increases with electron density. The arguments and modelling supporting the attribution to coupling between the graphene plasmon and

WS2 CB, leading to the formation of a polaron is mostly convincing.

However, I note that there is substantial quantitative differences in the theory calculations of the splitting between the main WS2 quasiparticle peak and the plasmon polaron peak (Fig 4e) and the experimental results (Fig 2e). These differences are not only in the magnitude of the splitting, which differ by a factor of over 10 in some cases, but also significantly in the rate of change with doping, which is much more significant in the experiments than is observed in the modelling. The authors should address this discrepancy.

There are also several details missing, and some aspects that are not well explained for the more general readership of Nature Communications, which should be addressed.

Specifically:

- The doping density is a key parameter and these values should be given with uncertainties.

- The authors state that the transfer of electrons from graphene to WS2 is “vanishingly small”, yet there is clearly some electron population in the WS2 CB. Presumably this comes from the graphene? Could this be quantified in the same manner that the doping density is determined for the bare WS2 sample? Being consistent and clear on this point would help the overall clarity of the manuscript.

- In describing the ARPES data in Fig 2 the authors note that there is significant band renormalization in the case of bare WS2, but no renormalization in the graphene/WS2 sample. This is not immediately obvious from the data in Fig 2. I would suggest EDC plots would help (as was done in [28]). On a related point, is the data for the bare WS2 the same data used in [28]? It looks remarkably similar and if so this should be made clear.

- In the modelling, the possible screening channels to the Coulomb interactions are important, and Fig 3 is used to depict these. However, more detailed explanation of these different mechanisms in the text are needed to aid understanding of what has been included.

Discovery of interlayer plasmon polaron in graphene/WS₂ heterostructures

Response to Reviewers

We thank the Reviewers for their careful evaluation of our manuscript. Below we provide a point-by-point reply to the comments of the Reviewers. Comments from the Reviewers are re-stated in green italicized text. Changes to the manuscript are marked in blue in the attached version for review.

Reviewer #1

“The authors conducted a combined experimental and theoretical investigation of the electronic and quasiparticle excitations in highly doped graphene/WS₂ heterostructures. Experiments are based on micro-focused angle-resolved photoemission spectroscopy (microARPES), whereas simulations are based on a model for the graphene WS₂ heterostructure.”

“The main finding of this manuscript consists in the discovery of a polaronic series in the conduction band bottom of monolayer WS₂ due to coupling with plasmonic excitations within graphene. The emergence of plasmonic polarons has been observed in the past in bulk and 2D semiconductors at sufficiently high doping concentration. However, to best of my knowledge this is the first instance for the observation of such phenomena in a hybrid heterostructure. Overall, this is a striking example of hybrid quasiparticle arising from many-body interactions involving electrons and plasmons at different ends of an hetero structure.”

“These results are based on high quality microARPES measurements. The authors have conducted a scan of different doping concentrations and revealed an increase of satellite energy with doping density. This analysis is the smoking-gun evidence for the plasmonic character of the satellite shake-off peaks. The theoretical analysis is based on the GW approximation for a simple model of the WS₂/graphene interface. This approach is below the state of the art, it describes a single plasmon satellites and it overestimates its energy. Even if the approach suffices to captures the key qualitative features of the experiments and to corroborate the interpretation of plasmon satellites, I believe that it would strengthen the manuscript to resort to a higher level theory, for instance by including vertex correction (as mentioned in the manuscript).”

“Overall the findings have sufficient novelty and originality to grant publication in Nature Communications. Before recommending the manuscript for publications, however, there is a final issue that the authors should address, to corroborate their interpretation of experimental data.”

Reply: We are delighted to see the positive response to our experimental observations and towards publication in Nature Communications. We especially like to thank the referee for clearly stating an important point, which we did not strengthened enough in our initial discussion: *“The emergence of plasmonic polarons has been observed in the past in bulk and 2D semiconductors at sufficiently high doping concentration.”*. In our case, the plasmon-polaron is observed in the WS₂ subsystem of our heterostructure, which is very little doped. The presence of clear and multiple shake-offs thus points towards the relevance of the heterostructure setup.

Concerning the level of the theoretical description we need to stress that our aim was and still is to qualitatively explain the trends and not to reach quantitative agreement. Nevertheless, we agree that exploring vertex corrections is interesting and possibly important. There are indeed already various methods available in the literature which allow for the investigation of such corrections to G0W0 theory. These methods include, but are not limited to, GW γ , dual approaches, Bethe-Salpeter and T-matrix based methods, and the so-called cumulant expansion methods. In the context of this work, especially the GW+cumulant expansion method is of interest [36, 38-41], as this approach is capable of producing higher order shake-off bands and to reduce the energy splitting between the quasi-particle and shakeoff peaks compared to G0W0 theory in 3D systems. Encouraged by the referee’s comment and our own curiosity, we implemented the retarded GW+cumulant expansion method as described by Kas et al. [36], which allowed us to gain deeper analytical insights into the origin and necessary characteristics of the driving plasmon in

the heterostructure, as we explain in detail below.

To support this discussion we added the following references to the manuscript:

- [36] J. J. Kas, J. J. Rehr, and L. Reining, “Cumulant expansion of the retarded one-electron Green function,” *Physical Review B* 90, 085112 (2014).
- [38] F. Aryasetiawan, L. Hedin, and K. Karlsson, “Multiple plasmon satellites in Na and Al spectral functions from ab initio cumulant expansion,” *Physical Review Letters* 77, 2268–2271 (1996).
- [39] Derek Vigil-Fowler, Steven G. Louie, and Johannes Lischner, “Dispersion and line shape of plasmon satellites in one, two, and three dimensions,” *Physical Review B* 93, 235446 (2016).
- [40] Gabriele Giuliani and Giovanni Vignale, *Quantum Theory of the Electron Liquid* (Cambridge University Press, Cambridge, 2005).
- [41] Mikhail I. Katsnelson, *Graphene: Carbon in Two Dimensions*, 1st ed. (Cambridge University Press, 2012).

“The authors attribute plasmons to the graphene layer. However, I would have expected a simpler and more intuitive picture, in which plasmons in the WS₂ layer are screened by graphene. Can the authors exclude this picture and unambiguously attribute plasmonic excitations to the graphene layer? A discussion of this aspect should be included in the manuscript.”

Reply: We thank the referee for raising this point and agree that the discussion needed to be enhanced. To this end, we used our new insights from the G0W0 + cumulant method and fully revised our discussion based on the analytical exploration of the cumulant method, from which we identified the following bosonic properties to be important:

- The boson energy at k_F (of WS₂), $\omega_{q=k_F}$, is directly proportional to the shakeoff energy splitting ΔE .
- The boson group velocity at k_F , $\frac{\partial\omega_q}{\partial q}|_{q=k_F}$, should be of similar magnitude to the WS₂ Fermi velocity for sharp shakeoff peaks.
- A dispersive and gapless bosonic dispersion relation results in finite spectral weight between shakeoff peaks.

All these characteristics are met for a plasmonic excitation in the graphene/WS₂ heterostructure. In detail, from this we understand that, for the shakeoff band to move over an energy scale of more than 100 meV upon K-doping, the plasmon energy at $q = k_F$ should change by the same amount.

If the relevant plasmonic excitation would be locally confined to the WS₂ layer, there are two effects which could cause such changes in the plasmonic energy: doping and screening. As for doping, from the ARPES data we learn that the WS₂ quasi-particle peak does not shift more than 3 meV upon the K-doping applied. Together with the fact that no shifts in the valence bands are observed, we can safely assume that the WS₂ occupation is not significantly altered upon K-doping. This excludes a change in the WS₂ doping as the cause for a WS₂-located plasmon energy shift. As for screening, static screening from the graphene layer would indeed change the energy scale of the WS₂-located plasmon and would be sensitive to the doping of graphene. However, already from a Thomas-Fermi screening model we understand that as the doping of graphene is increased, the screening increases, such that the WS₂ plasmon energy decreases with enhanced K-doping. This is opposite to the trend which is observed experimentally, such that we also exclude this mechanism as the cause for the shifting shakeoff peaks. Furthermore, the WS₂ plasmon dispersion lies within the electron hole continuum of graphene and is therefore exposed to Landau damping in form of single particle electron-hole excitations within the graphene layers. As a consequence, the electron-plasmon interaction is also reduced. Taking all of this together, the WS₂ plasmon is unlikely to be able to cause clear shakeoff peaks as sharp as observed in the ARPES data with the correct magnitudes and trends.

Unambiguously attributing the shakeoff bands to the graphene plasmon is, however, challenging as well due to the approximations in our model. Nevertheless, from the analytical exploration described above we understand that the plasmon energy at $q = k_F$ (with k_F the Fermi vector of WS₂) should change on the order of 50 to 150 meV upon graphene doping, should be dispersive and should have a group velocity which is of similar magnitude as the WS₂ Fermi velocity. This suggests a 2D plasmon mode that significantly changes in energy as the graphene layer is doped. Therefore, the graphene plasmon seems like a natural candidate, even though its energy scale might be too large and its group velocity might be too steep for the formation of sharp shakeoff bands on the 50 - 150 meV energy range below the conduction band minimum at the experimentally verified low WS₂ doping (and thus small k_F). To derive an unambiguous final answer to this, we would however need to go beyond the current model by taking all atomic degrees of freedom of the heterostructure into account and reconsider the G0W0 starting point of the our cumulant expansion, which is beyond the scope of our manuscript. Especially, as there might be further explanations, such as

hybridized modes between the graphene plasmon and a phonon mode which could have the correct properties.

Nevertheless, our presented experimental data on the comparison between doped WS₂ and doped graphene/WS₂ heterostructures clearly proves the relevance of the heterostructure setup and the non-local Coulomb interaction within it. Keeping in mind the low WS₂ doping level and the significantly suppressed hybridization between WS₂ and graphene, the driving boson for the observed WS₂-located plasmon-polarons must therefore arise from the Coulomb interactions between the two subsystems. This might be either an interlayer plasmon mode or a graphene-located plasmon.

To clarify this we revised our theoretical discussion. In our new analysis, we assume a plasmon dispersion of the shape $\omega_q = \sqrt{2ve^2q/\pi\epsilon_q}$, with v a doping-tunable parameter and ϵ_q a non-local background dielectric function taking into account the layered structure, and an electron-plasmon coupling $a_q^2 = \omega_q U_q/2$, with U_q the bare Coulomb interaction, as it arises, e.g., from an interlayer plasmon mode. We show that this plasmon mode can quantitatively reproduce the ARPES spectra by tuning the doping via v and, due to the cumulant vertex correction, we also find higher shake-offs. To support this discussion we replaced Fig. 4 with the following figure and caption (and moved it up in the paper, such that it is now Fig. 3):

Fig. 3: Theoretical results. **a-b**, The plasmon dispersion ω_q and electron-plasmon coupling a_q^2 , respectively, for various v . The vertical dotted line denotes $q = k_F$. **c-e**, EDCs of the WS₂ normal state spectral function in G_0W_0 theory (green dashed) and G_0W_0+C theory (red solid) at \bar{K} for a variety of v . The vertical dotted black lines denote $\omega = -\mu_{\text{WS}_2} - n(\omega_{q=k_F} - \mu_{\text{WS}_2})$, for $n = 0$ to 4. **f**, Energy splitting ΔE between the WS₂ CBM and the first shakeoff band as a function of v , in G_0W_0 theory (green dashed) and G_0W_0+C theory (red solid). The black dotted line denotes $\omega_{q=k_F} - \mu_{\text{WS}_2}$, and the gray horizontal lines denote the experimentally measured ΔE .

Reviewer #2

“Ulstrup and coauthors investigate the presence of interlayer plasmon polarons in graphene/WS₂ heterostructures. In particular, the authors observe so-called shake-off bands in the ARPES signal of WS₂ conduction band electrons. The qualitative agreement between the measured doping dependence of the shake-off bands and the theoretical model indicates that the observed signatures originate from WS₂ conduction band electrons coupled to graphene plasmons. The findings are clearly presented and the conclusions are well supported. Overall, the work provides an important step forward in the context of heterostructures formed by graphene and transition metal dichalcogenides, which are

currently relevant for fundamental studies and potential applications. While I think this work should be published in Nature Communications, I ask the authors to address my comments before I can make a definitive recommendation.”

Reply: We are happy to see the positive recommendation and thank the Reviewer for the revision suggestions that have improved the manuscript.

“1. The lack of hybridization between the Dirac cone and the WS_2 CBM at K , which is a key aspect for the realization of the interlayer plasmon polaron, was already reported in other works [PRL 127, 276401 (2021); 2D Mater. 10 (2023) 035025]. The authors are aware of at least one of these articles, which they cite in the supplementary information but not in the main manuscript. I ask the authors to refer to these works in the manuscript.”

Reply: We fully agree with the Reviewer and have now included these citations ([30] and [31]) in the main manuscript in the following segment on page 6 of the revision:

“Density functional theory (DFT) calculations for the K/graphene/ WS_2 heterostructure (see Methods and Supplementary Figs. 4-5) confirm the experimental results which show that the graphene Dirac bands do not strongly hybridize with the WS_2 CBM at K, in line with previous reports [30, 31]. ”

“2. The authors consider the structure $hBN/WS_2/graphene/dopants$. Out of curiosity, how are the results expected to change if the dopants are located on top of WS_2 , e.g. in an $hBN/graphene/WS_2/dopants$ structure?”

Reply: We agree that such experiments would also be interesting to pursue. In that case, one would anticipate strong doping in the WS_2 from the adsorbed alkalis and much less doping in the underlying graphene, however, it is an open question how the screening from the underlying graphene, and internally in WS_2 from the larger density of free carriers, would impact the interlayer interactions in this case. As shown in Fig. 2 of our manuscript, appreciable graphene-doping ($> 4e13cm^{-2}$) is required for the polaron to be observable, which we understand as an optimal doping regime to reach a small WS_2 doping accompanied by inter-layer plasmon modes with the necessary analytic properties in form of plasmon energies and group velocities at the k_F of WS_2 . It is questionable whether a similar optimal balance can be reached by direct alkali adsorption on top of WS_2 on graphene. This would need be experimentally investigated, which is outside the scope of the present manuscript.

“3. It is not clear from Fig.1e that the K valley occupation in WS_2 is small, as the authors state. It makes sense that it should be small, but I do not see how the authors reach the conclusion from Fig.1e alone. This should be better explained. It could be helpful, or at least improve transparency, to show in the extended data the conduction band signal of both the WS_2 K valley and the graphene Dirac cone together in a single plot.”

Reply: We thank the Reviewer for pointing this out and helping to improve clarity of the manuscript. We have followed the suggestion and added a new Fig. S3 in the Supplementary Information that plots the doped graphene and WS_2 dispersions along the relevant k-routes in the BZs of graphene and WS_2 , highlighting the significant filling of graphene compared to the WS_2 CBM. The main text has been extended accordingly on page 5 with the following new sentence:

“The strongly doped graphene is accompanied by a relatively small occupation in the WS_2 CBM (see ARPES spectra of doped WS_2 and graphene in Supplementary Fig. 3).”

The new Fig. S3 with caption is reproduced on the next page.

“4. While Fig.2c is very clear, its connection with Fig.2d is not really so, given that only one line and not the cone is resolved in Fig.2d. Why is only one line seen in Fig.2d? And how can the Fermi momentum be extracted from this, i.e. how is the center of the cone extracted from the data? This needs to be more clearly explained.”

Reply: We agree with the Reviewer that the presentation of this part of the analysis was not sufficiently clear for the reader. We have now clearly marked k_F in both panels 2(c) and 2(d) and added the following explanation on page 6 that addresses the comment of the Reviewer:

“The graphene wave vector k_F , illustrated with the Dirac cone in Fig. 2(c), is extracted from ARPES cuts through the center of the graphene Dirac cone at \bar{K}_C , as shown for doped graphene on WS_2 in Fig. 2(d). The Fermi

FIG. S3. ARPES data for doped graphene/WS₂. The black and blue Brillouin zones (BZs) correspond to graphene and WS₂, respectively. The rotation between the BZs corresponds to a twist angle of $(18.1 \pm 0.3)^\circ$ between graphene and WS₂. The red line demarcates the direction of the ARPES cut. The graphene carrier concentration is $(5.2 \pm 0.1) \cdot 10^{13} \text{ cm}^{-2}$. The Dirac energy E_D and WS₂ CBM are indicated by arrows.

momentum is then obtained from an MDC fit at E_F and given as the difference in k between the MDC peak position and \bar{K}_G . Note that \bar{K}_G is determined by mapping the (E, k_x, k_y) -dependent ARPES intensity around the Dirac cone. One of the Dirac cone branches is suppressed in Fig. 2(d) because of strong photoemission matrix element effects along this particular cut, which is taken along the so-called dark corridor [32].”

New Ref [32]: I. Gierz et al., Phys. Rev. B 83, 121408 (2011)

“5. I have a few questions regarding the shake-off bands. Addressing these points in the manuscript or supplementary information could be helpful for future works trying to understand this phenomenon better.

1. Does the number of shake-off bands change with doping?”

Reply: The ability to experimentally observe shake-offs in our conditions depends on doping. We have extended the discussion of Fig. 2 on pages 6-7 with the following paragraph:

”The EDC analysis of the shake-off bands as a function of graphene doping reveals the energy separation between shake-off bands increases from $(50 \pm 8) \text{ meV}$ to $(141 \pm 18) \text{ meV}$ and that the increase is proportional to $\sqrt{n_G - n_0}$, as shown in Fig. 2(e), while the WS₂ CBM binding energy, and thus doping level, approximately stays constant. Note that a minimum carrier density in graphene of $n_0 = (4.1 \pm 0.1) \cdot 10^{13} \text{ cm}^{-2}$ is required for the WS₂ CBM to become occupied and thereby make the shake-off bands observable. The EDC fits in Fig. 2(b) demonstrate that the shake-off band intensity relative to the main quasiparticle peak diminishes with doping in line with our theoretical analysis below. Combined with the diminishing intensity of shake-offs towards higher binding energies, this reduces the number of shake-off bands we can observe with increasing doping.”

“2. Is the dispersion of the shake-off bands expected to be the same as that of the CBM?”

Reply: For three-dimensional systems, plasmon polaron shakeoff bands are known to follow the dispersion of the CBM [38-40]. For two-dimensional systems, we understand from the G_0W_0 +cumulant method that, due to the dispersion of the 2D plasmon, the plasmon polaron shakeoff band does not follow the dispersion of the CBM. Instead, the shakeoff band merges with the conduction band at the Fermi wavevector, which is in agreement with previous works on two-dimensional plasmon polarons [40,41].

To support this discussion we added the following references to the manuscript:

[38] F. Aryasetiawan, L. Hedin, and K. Karlsson, “Multiple plasmon satellites in Na and Al spectral functions from

ab initio cumulant expansion,” Physical Review Letters 77, 2268–2271 (1996).

[39] Derek Vigil-Fowler, Steven G. Louie, and Johannes Lischner, “Dispersion and line shape of plasmon satellites in one, two, and three dimensions,” Physical Review B 93, 235446 (2016).

“3. Why does the energy and momentum width of the CBM signal decrease with the presence of graphene, and is this reproduced by the theoretical model? Related to this, can an effective mass and temperature be extracted from the energy and momentum width of the CBM signal?”

Reply: With K doping the occupation of the WS₂ K valley (where the CBM is situated) is vastly different with and without graphene. Without graphene WS₂ is much more doped by K than it is with the graphene layer in between. Correspondingly k_F in WS₂ is smaller (larger) with(out) graphene, which explains the change in the “momentum width”. Changes to the “energy width” are likely resulting from the many-body broadening effects, e.g., due to the electron-plasmon coupling, which is, again, considerably different with and without the graphene layer. Without graphene the corresponding plasmon is the one within WS₂ (at large doping), while with graphene, as we also argue above, the relevant plasmon is an interlayer plasmon. These tendencies are reproduced by our revised theory, where we find reduced CBM intensity due to interactions with the low-energy interlayer plasmon. As a result, fitting an effective temperature would neglect these additional (non-temperature related) broadening effects. Effective mass fits are, theoretically, still possible for the quasi-particle dispersion, but the experimental resolution (as a result of the small k_F with and without graphene) does not permit such an analysis.

“4. Why does the shake-off band intensity relative to the main quasiparticle peak diminish with doping? Perhaps the theoretical model can provide insights into this.”

Reply: As the doping increases, the plasmon energy $\omega_{q=k_F}$ increases as well. From the GW+cumulant method we understand that the intensity of the first shakeoff band is proportional to $1/(\omega_{q=k_F} - \mu_{\text{WS}_2})^2$, such that the shakeoff band intensity reduces with increased plasmon energy. In a more intuitive picture, we can understand the shakeoff bands as electrons dressed by plasmonic excitations. With increasing doping and due to the dispersive character of 2D plasmons, the relevant plasmonic energies (or plasmonic “band width”) involved in this process are getting larger (the band-width gets wider) such that also the resulting shakeoff gets broadened. This is further reflected in the theoretical spectral function by finite spectral weight between the quasiparticle and the shakeoff peaks.

“5. I understand that the theory considered can only predict a single shake-off band. However, I wonder if it is expected that additional sidebands will display the same energy separation as seen in Fig.2b. Related to this, what is the physical meaning of the energetic separation between the shake-off bands? For phonon polarons it is given by the phonon energy, if I am not mistaken. Can it be related to some characteristic energy of the graphene plasmons here?”

Reply: The GW+cumulant method now reproduces further shakeoff bands and gives us a clear understanding on the physical meaning of the separation between the shakeoff bands. In short, if the WS₂ occupation is low, one can understand the separation between shakeoff bands to be equal to $\Delta E = \omega_{q=k_F} - \mu_{\text{WS}_2}$, with $\omega_{q=k_F}$ the plasmon energy at the k_F of WS₂ and μ_{WS_2} the WS₂ chemical potential. From this we understand that the characteristic plasmon energy is the plasmon dispersion evaluated at the WS₂ k_F . In the revised Fig. 3 (previously Fig. 4) we also indicate this energy with grey vertical dashed lines.

“6. Regarding the sentence “the doping dependence of the line shape of the shake-off bands is consistent with graphene plasmon excitations coupling to the WS₂ conduction electrons”, I ask the authors to be more precise: how does the line shape depend on doping and how is this consistent with the mentioned effect?”

Reply: We agree with the Reviewer that this sentence was somewhat confusing. We merely meant to summarize that the energy-separation of shake-off bands is doping dependent to explain the plasmon polaron picture. We replaced this sentence with the following on page 8:

”Taken together with the significant doping dependence of the energy separation between shake-off bands, this suggests that the observed feature is an interlayer plasmon polaron...”

“7. Where is the plasmon polaron band in Fig 4c? The signatures below the dotted lines are a bit complicated, resembling a sort of triangular shape. Is the polaron band the upper part of this “triangle? Then what are the sides?”

Please explain more clearly and indicate where the polaron band is in the plot. Indicating the energy separation between the dotted lines and the plasmon polaron band in the plot would help.”

Reply: We agree with the referee that the previous panel Fig. 4c was not very clear. For the revised theoretical discussion, we decided to show only energy distribution curves at \bar{K} of WS_2 in the (now) Fig. 3c-e. In these EDCs the main quasiparticle peak are clearly visible next to the (multiple) shakeoff peaks.

“8. Can the authors show that the theoretical trend shown in Fig.4e also follows a square root dependence as they state in the text?”

Reply: As we explain above, the energy splitting in the GW + cumulant theory is given by $\Delta E = \omega_{q=k_F}(n_G) - \mu_{\text{WS}_2}$. For a 2D plasmon the assumption $\Delta E \propto \sqrt{n_G}$ is thus adequate.

“9. Can the authors foresee some of the expected consequences of the observed interlayer plasmon polarons? In particular, would they weaken or enhance the electron-electron interactions in WS_2 ? From this one could qualitatively predict how the stability of correlated many-body phases such as Wigner crystals would be affected.”

Reply: The reviewer raises a very interesting point. It is true that now that we have proven that the heterostructure induces many-body effects beyond the individual constituents, possible impacts to many-body ground-states or excitations should be investigated further. However, we like to stress that our analysis clearly shows, that the effective electron-electron interaction in WS_2 is vastly affected by the interface in terms of its momentum and frequency dependence. Both can have non-trivial impacts to many-body properties, which deserve further explorations in the future.

Reviewer #3

“The manuscript by Ulstrup et al report the observation of a new many body quasiparticle - an interlayer plasmon polaron. Plasmon polarons have been reported previously, but the observation of the coupling occurring across two different monolayer materials is a significant new discovery. There is much interest in low-energy bosonic modes, such as plasmon polarons, for realising and/or stabilising different strongly correlated electron phases (eg superconductivity) and novel quantum phases. In this particular case, the ability to couple the plasmon in electron doped graphene to the bandstructure of a monolayer semiconductor provides a new avenue for controlling the properties of such materials.”

“The main experimental evidence is the ARPES data showing shake off bands below the WS_2 conduction band, with energy splitting that increases with electron density. The arguments and modelling supporting the attribution to coupling between the graphene plasmon and WS_2 CB, leading to the formation of a polaron is mostly convincing. However, I note that there is substantial quantitative differences in the theory calculations of the splitting between the main WS_2 quasiparticle peak and the plasmon polaron peak (Fig 4e) and the experimental results (Fig 2e). These differences are not only in the magnitude of the splitting, which differ by a factor of over 10 in some cases, but also significantly in the rate of change with doping, which is much more significant in the experiments than is observed in the modelling. The authors should address this discrepancy.”

Reply: We are happy to see that our experimental approach and observation of a plasmon polaron is well-received by the Reviewer. Concerning the theory, we would first like to stress that the aim never was to reach quantitative agreement. As we wrote in the manuscript, we attributed the discrepancies between theory and experiment to the simplicity of our model, as well as to so-called ‘vertex corrections’, which were missing in the G0W0 theory.

Upon request by Reviewer 1, we implemented and applied a higher level theory, the so-called G0W0 + cumulant method, which gives us a better understanding of the plasmon properties required for the formation of shakeoff bands as seen in the ARPES spectra. Based on this, we understand that the inherent WS_2 plasmon mode cannot be responsible for the observed behaviour. In the revised theoretical discussion, we now show that, within the G_0W_0 +cumulant approach, a 2D plasmon (either from the interface or hosted within graphene) can indeed yield quantitative agreement.

“There are also several details missing, and some aspects that are not well explained for the more general readership

of Nature Communications, which should be addressed. Specifically:”

Reply: We thank the Reviewer for bringing these missing details to our attention and helping us to improve the manuscript.

“- The doping density is a key parameter and these values should be given with uncertainties.”

Reply: We have now added error bars to all experimentally determined doping values. For the values reported in Fig. 2a, we state the error bars in the caption.

“- The authors state that the transfer of electrons from graphene to WS_2 in “vanishingly small”, yet there is clearly some electron population in the WS_2 CB. Presumably this comes from the graphene? Could this be quantified in the same manner that the doping density is determined for the bare WS_2 sample? Being consistent and clear on this point would help the overall clarity of the manuscript.”

Reply: The Reviewer raises an important point, and it should be clearly explained why we are not reporting the WS_2 doping for graphene/ WS_2 . The explanation, which has been added in the Methods section on page 12, is as follows:

“It is not possible to determine the doping of WS_2 under graphene in a similar way as for bare WS_2 because the CBM remains extremely flat and pinned at E_F , preventing any meaningful extraction of a Luttinger area. We therefore only report the graphene doping level for graphene/ WS_2 heterostructures, which can be reliably extracted as described above.”

We further like to stress, as also mentioned above, that there are no experimental signs for a change in the WS_2 doping. Thus, the only important fact for the subsequent analysis is that the WS_2 doping is small. We do believe that this is clearly visible by the eye and does not require a numerical quantification.

“- In describing the ARPES data in Fig 2 the authors note that there is significant band renormalization in the case of bare WS_2 , but no renormalization in the graphene/ WS_2 sample. This is not immediately obvious from the data in Fig 2. I would suggest EDC plots would help (as was done in [28]). On a related point, is the data for the bare WS_2 the same data used in [28]? It looks remarkably similar and if so this should be made clear.”

Reply: We agree with the Reviewer and have modified Fig. S1 to include an analysis of the VBM region as well (i.e. we have extended the energy range of the plots and included these parts in the analysis). This shows the expected spin-orbit splitting of 0.43 eV at K of WS_2 in doped graphene/ WS_2 , while a strongly renormalized splitting of 0.67 eV is seen in the VBM at K of bare doped WS_2 . The modified Fig. S1 is reproduced here with its caption:

”FIG. S1. Extraction of band gaps from ARPES. a, Energy distribution curves (EDCs) at K in the heavily doped situations shown in Figs. 1(c) and 1(e) of the main paper. The black curves represent fits to a modelled spectral

function with several Lorentzian peaks on a linear background and including a Fermi-Dirac cut-off. b-c, ARPES spectra around K, displaying the VBM and CBM of SL WS₂ (b) with and (c) without a graphene overlayer. VBM and CBM peak positions obtained from the EDC analysis in (a) and the resulting energy differences are stated in units of eV. The error bars are ± 0.02 eV.”

Concerning the data for bare WS₂, it is not the same as used in our previous work. It was taken under exactly the same conditions at the same beamline and during the same beamtime (i.e. same sample orientation, same photon energy, same doser etc.).

“- In the modelling, the possible screening channels to the Coulomb interactions are important, and Fig 3 is used to depict these. However, more detailed explanation of these different mechanisms in the text are needed to aid understanding of what has been included.”

Reply: In the revised manuscript we included an extended discussion on the possible screening channels. Note that this has been moved to Fig. 4.

REVIEWERS' COMMENTS

Reviewer #1 (Remarks to the Author):

The authors have addressed all my comments in the revised version of the manuscript, and I therefore recommend the manuscript for publication in Nature Communication.

Reviewer #2 (Remarks to the Author):

The authors have clarified all my questions and have even improved the theoretical description of the phenomenon. I strongly recommend the article to be published in Nature Communications.

Reviewer #3 (Remarks to the Author):

In responding to the referee comments the authors have significantly improved the manuscript. They have appropriately clarified the issues raised, and the extension of their modelling to include the cumulant expansion has added significantly to the robustness of the interpretation and understanding.

Combined with the quality and significance of the experimental results identified in the initial review, this now makes for an excellent paper, worthy of publication in Nature Communications.